# Direct true lumen cannulation in type A acute aortic dissection: A review of an 11 years' experience

Hazem El Beyrouti[1]*, Daniel-Sebastian Dohle[1], Mohammad Bashar Izzat[2], Lena Brendel[1], Philipp Pfeiffer[1], Christian-Friedrich Vahl[1]

1 Department of Cardiothoracic and Vascular Surgery, Medical Center of the Johannes Gutenberg University Mainz, Mainz, Germany, 2 Department of Surgery, Faculty of Medicine, Damascus University, Damascus, Syria

* hbeyrouti@gmail.com

**Data Availability Statement:** All relevant data are within the paper and its Supporting Information files.

## Abstract

### Objectives

Direct true lumen cannulation (DTLC) of the aorta is an alternative cardiopulmonary bypass cannulation technique in the context of type A acute aortic dissection (A-AAD). DTLC has been reported to be effective in restoring adequate perfusion to jeopardized organs. This study reports and compares operative outcomes with DTLC or alternative cannulation techniques in a large cohort of patients with A-AAD.

### Methods

All patients who underwent surgery for A-AAD between January 2006 and January 2017 in Mainz university hospital were reviewed. The choice of cannulation technique was left to the operating surgeon, however DTLC was our preference in patients who were in state of shock or showed signs of tamponade or hypoperfusion, in cases of potential cerebral malperfusion, as well as in patients who were under resuscitation.

### Results

A total of 528 patients (63% males, mean age 64±13.8 years) underwent emergency surgery for A-AAD. The DTLC technique was used in 52.4% of patients. The DTLC group of patients had worse clinical status at the time of presentation with more shock, tamponade, true lumen collapse, cerebral and other malperfusion states. New neurologic events were diagnosed in around 8% of patients in each group following surgery, but there was a trend for quicker neurological recovery in the DTLC-group. Early mortality rates, short-term and long-term survival rates did not differ between the two groups.

### Conclusions

DTLC is a safe cannulation technique that enables effective antegrade true lumen perfusion in complicated A-AAD scenarios, and is an advantageous addition to the aortic surgeons' armamentarium.

**Funding:** The author(s) received no specific funding for this work

**Competing interests:** The authors have declared that no competing interests exist.

## Introduction

Rapid cannulation of the true lumen and restoration of adequate perfusion to jeopardized organs is necessary for the success of surgery for type A acute aortic dissection (A-AAD), especially in the presence of pre-operative hemodynamic instability and distal organ malperfusion. While femoral arterial cannulation with retrograde perfusion with cardiopulmonary bypass (CPB) has been used extensively in the management of A-AAD [1], antegrade perfusion has been shown to more effective in preventing malperfusion, embolization and organ dysfunction [2].

Following the first description by Borst *et al* [3], several groups reported their experiences with direct true lumen cannulation (DTLC) in the context of A-AAD [4–6]. In this paper, we report our experience with the use of the DTLC techniques over an 11 years' period, and compare outcomes of DTLC with alternative cannulation techniques in a large cohort of patients with A-AAD.

## Patients and methods

All patients who were operated on for A-AAD in Mainz university hospital between January 2006 and January 2017 were identified in our institutional database using the International Classification of Diseases codes (ICD-10), and their records were retrieved. Approval from the responsible ethics committee of the State Medical Association of Rheinland Pfalz was obtained for data analysis (2018-13574-Epidemiologie), Mainz, Germany. Informed patient consent was waived for this retrospective study.

## Procedural details

The diagnosis of A-AAD was established using CT angiography, echocardiography or angiography. Upon establishing the diagnosis of A-AAD, patients were transferred to the operating room directly. Intra-operative monitoring techniques included bilateral cerebral oxygen saturations (INVOS Somanetics, Troy, MI), unilateral femoral and bilateral radial arterial blood pressures, bladder and rectal temperatures, and transesophageal echocardiography.

The choice of cannulation technique was based mainly on the experience of the operating surgeon, as well as on the extent of the planned aortic replacement. The DTLC was previously described [7]. Briefly, a median sternotomy was performed, the pericardium was opened, and heparin was administered. The diseased aorta was carefully mobilized off-the main pulmonary trunk and encircled with two surgical tapes. Next, a two-stage venous cannula was inserted in the right atrium and connected to the CPB venous line. Under pure oxygen ventilation, the patient was placed in a steep head-down position, and venous blood was drained freely into the venous reservoir of the CPB circuit until the systolic blood pressure dropped to approximately 30 mmHg. With the diseased aorta thus decompressed, the ascending aorta was opened halfway between the sino-tubular junction and the origin of the innominate artery, and the true lumen was identified. A 24F arterial cannula with straight tip (FLK A34 D24, Free life medical GmbH, Aachen), which was already connected to CPB arterial line, was introduced into the true lumen under direct vision, and was de-aired passively by allowing the level of blood to rise slowly in the aorta and actively by low-flow perfusion from the CPB machine. Low-flow forward CPB perfusion was established, and the surgical tapes were gently snared around the aorta to secure the arterial cannula in place. Finally, full-CPB perfusion was established. Overall, the low-pressure period lasted less 90 seconds (S1 Video).

While cooling the patient to the designated core temperature, selective antegrade and/or retrograde cold cardioplegia were employed for myocardial protection. For supra-coronary replacement of the ascending aorta, the sino-tubular junction was reconstructed and was

anastomosed to an adequately sized Dacron tubular graft (Braun Aesculap, Tuttlingen, Germany). When the aortic root was considered unsalvageable, a David-style aortic valve re-implantation procedure was carried out when possible, or a standard aortic root replacement with a valved-conduit was carried out.

Upon reaching the targeted rectal temperature (mean 23±6˚C), circulatory arrest was initiated. The surgical tapes around the aorta were released, the arterial cannula was removed and the aortic arch was inspected. The distal aortic anastomosis was performed, reinforced with a Teflon felt strip (polytetrafluoroethylene felt, Impra, Tempe, AZ). If the intimal tear was located within the aortic arch, the dissected arch was partially or totally replaced. Afterwards, the aortic cannula was re-introduced into the aortic graft through a stab incision, and antegrade perfusion was restored after venting the graft. Rewarming and weaning from CPB were performed in the standard fashion. The stab incision in the aortic graft was closed with a pledgeted suture after removal of the arterial cannula.

## Statistical analysis

Data were collected prospectively, supported by our database for thoracic aortic operations, and were analyzed retrospectively. Statistical computations were performed using the SPSS 22.0 (SPSS Statistics for Macintosh, Version 22.0. Armonk, NY: IBM Corp.) and wizard pro data analysis version 1.9.7 (Evan Miller, Chicago, IL). All frequency data are presented as percentages and all continuous data as the mean ± standard deviation. Normal distribution of continuous variables was validated using the Shapiro–Wilk test. In the case of 2 groups, continuous variables were compared by the *t*-test. If the assumption did not hold, the Wilcoxon rank sum test was used. Nominal variables were compared using the Fisher's exact test. Figure one was calculated with GraphPad prism version 7.0a for Mac (GraphPad Software, La Jolla, CA, USA) and survival was assessed with the use of the Kaplan–Meier method and compared with the use of the log-rank test. The alpha level was set at 0.05 for statistical significance.

## Results

### Demographics and preoperative status

Between January 2006 and January 2017, a total of 528 patients (63% males, mean age 64±13.8 years) underwent emergency surgery for A-AAD at our department in Mainz University Hospital. Two-thirds of these cases were classified as type I DeBakey dissections, and one third as type II DeBakey dissections (Table 1). The diagnosis of A-AAD was established using CT angiography in 84% of patients, echocardiography in 14%, and angiography in 2% of patients.

Upon presentation, 20.49% of patients were in the state of shock (defined as systolic hypotension <80 mmHg accompanied with organ hypoperfusion despite resuscitative therapy), and 16.7% of patients showed signs of cardiac tamponade (defined as low cardiac output due to a compressing pericardial effusion). Signs of cerebral malperfusion were also detected in 11.55% patients, and 12.69% of patients were artificially ventilated. Thirty patients (5.69%) had undergone cardiopulmonary resuscitation or were under resuscitation. Patients' demographics, comorbidities and the pre-operative clinical status according to the Penn classification are summarized in Table 1.

As stated above, the choice of cannulation technique was based mainly on the experience of the operating surgeon and on the extent of the planned aortic replacement. By and large, DTLC was our preferred strategy in patients who were in state of shock or showed signs of tamponade or hypoperfusion, and in cases of potential cerebral malperfusion (e.g. dissection of the brachiocephalic trunk) as well as in patients who were under resuscitation. Overall, the

**Table 1. Patients demographics, comorbidities and clinical status at presentation.**

| | all patients | DTLC group | non-DTLC group | *p* |
|---|---|---|---|---|
| no. of patients | 528 | 277 | 251 | - |
| Age (years) | 64±13.8 | 63±13.8 | 64.8±13.7 | 0.276 |
| male gender | 334 (63.26%) | 182 (61.07%) | 152 (66.08%) | 0.236 |
| BMI | 27.4±5.2 | 27.6±5.3 | 27.3±5.2 | 0.559 |
| COPD | 35 (6.63%) | 18 (6.04%) | 17 (7.89%) | 0.336 |
| AHT | 384 (72.73%) | 221 (74.16%) | 163 (70.87%) | 0.4 |
| DM | 57 (10.8) | 33 (11.07%) | 24 (10.44%) | 0.815 |
| smoking | 106 (20.08%) | 57 (18.13%) | 48 (21.30%) | 0.336 |
| CAD | 95 (17.99%) | 36 (12.08%) | 59 (25.65%) | <0.001 |
| clinical status | | | | |
| AI | 397 (75.19%) | 205 (74.01%) | 192 (76.49%) | 0.837 |
| TLC | 108 (20.49%) | 73 (24.5%) | 35 (15.28%) | 0.009 |
| tamponade | 88 (16.7%) | 61 (20.47%) | 27 (11.78%) | 0.008 |
| shock | 108 (20.49%) | 78 (26.17%) | 30 (13.1%) | <0.001 |
| cerebral malperfusion | 61 (11.55%) | 43 (15.52%) | 18 (7.17%) | 0.003 |
| other malperfusion | 168 (31.82%) | 101 (36.46%) | 67 (26.69%) | 0.016 |
| CPR | 30 (5.69%) | 19 (6.38%) | 11 (4.8%) | 0.440 |
| artifical ventilation | 67 (12.69%) | 43 (15.52%) | 24 (9.56%) | 0.040 |
| DeBakey Classification | | | | |
| type I | 353 (66.86%) | 201 (72.5%) | 152 (60.5%) | 0.003 |
| type II | 175 (33.14%) | 76 (27.5%) | 99 (39.5%) | 0.003 |
| Penn Classification | | | | |
| class Aa | 273 (52%) | 122 (44.04%) | 151 (60.16%) | 0.002 |
| class Ab | 124 (23%) | 75 (27.08%) | 49 (19.52%) | 0.002 |
| class Ac | 62 (12%) | 35 (12.64%) | 27 (10.76%) | 0.002 |
| class Abc | 69 (13%) | 45 (16.25%) | 24 (9.56%) | 0.002 |

Data is presented as mean±SD or as number (%) as appropriate.

*BMI*: body mass index, *COPD*: chronic obstructive pulmonary disease, *AHT*: arterial hypertension, *DM*: diabetes mellitus, *CAD*: coronary artery disease, *TLC*: true lumen collapse, *CPR*: cardiopulmonary resuscitation, *HP*: hypoperfusion, *AI*: aortic insufficiency, *Penn class Aa*: no shock or hypoperfusion, *Penn class Ab*: hypoperfusion, *Penn class Ac*: shock, *Penn class Abc*: shock and hypoperfusion.

DTLC technique was used in 52.4% of patients (n = 277, the DTLC group), while alternative cannulation techniques were used in 47.6% of patients (n = 251, the non-DTLC group); including femoral (25%), aortic arch (14%), ascending aortic (4.7%), axillary (1.9%), brachiocephalic (0.4%), carotid (0.2%), left atrial (0.6%) and multiple cannulations (0.8%).

Patients' demographics were similar in both groups. Coronary artery disease was less frequently encountered in the DTLC group compared to the non-DTLC group (12% vs. 25.65%; *p*<0.001). Notably, the DTLC group included significantly more patients with type I DeBakey dissections than the non-DTLC group (72.5% vs. 60.5%; *p*<0.001). When categorized according to the Penn classification of pre-operative clinical status, significantly more patients from the DTLC-group were in the class Ab (27.08% vs. 19.52%), class Ac (12.64% vs. 10.76%) and class Abc (16.25% vs. 9.56%) categories (Fig 1; all *p* = 0.002). This was reflected by worse clinical status at the time of presentation of patients in the DTLC group, with more shock (26.17% vs. 13.1%; *p*<0.001), tamponade (20.47% vs. 11.78%; *p* = 0.008), true lumen collapse (24.5% vs. 15.28%; *p* = 0.009), cerebral malperfusion (15.52% vs 7.17%; *p* = 0.003) and other malperfusion states (36.46% vs 26.69%; *p* = 0.016) (Table 1).

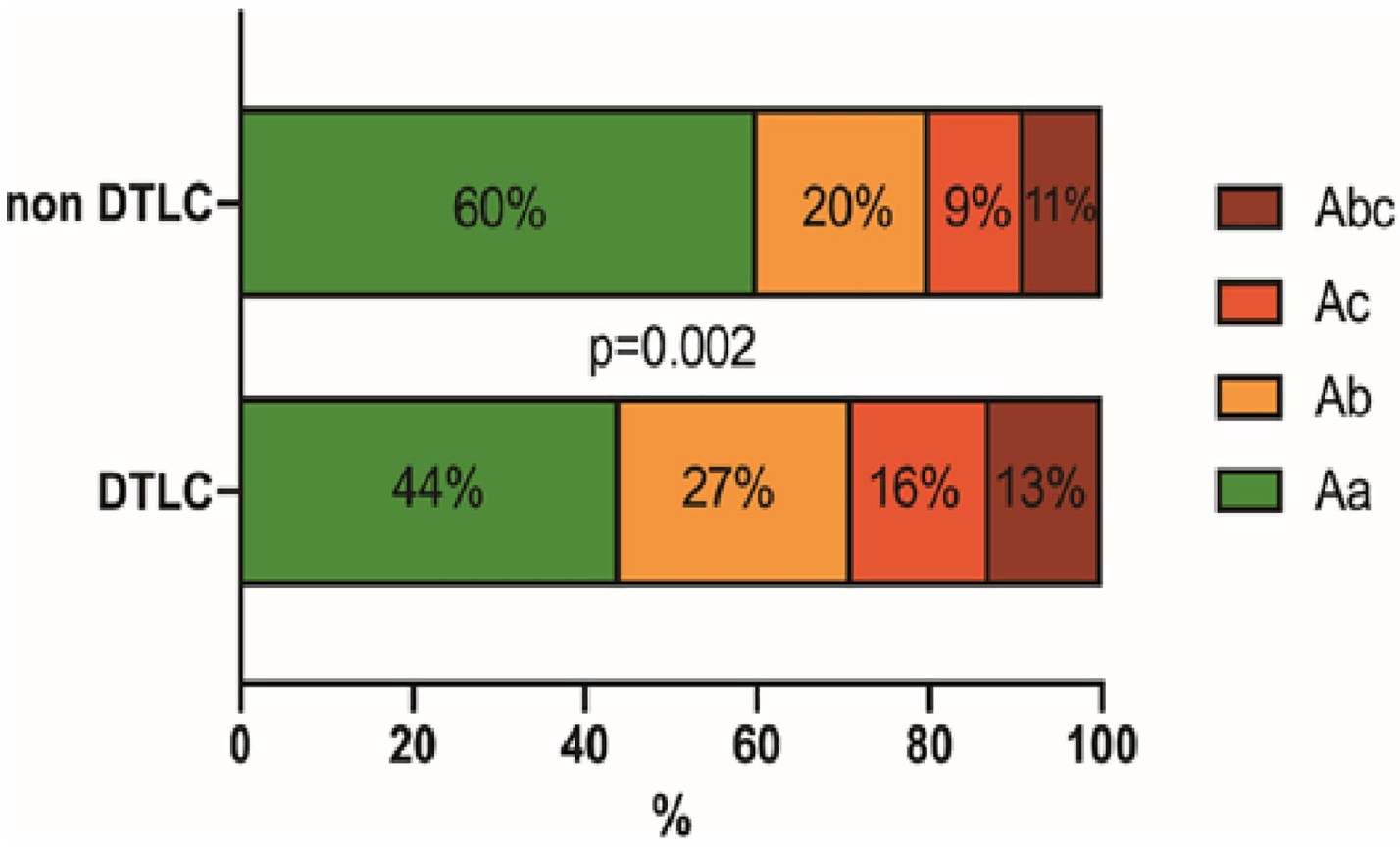

**Fig 1. Distribution of preoperative clinical status according to the Penn classification for patients in DTLC and non-DTLC groups.**

## Operative data

Supra-coronary replacements of the ascending aorta were performed less in the DTLC group compared to the non-DTLC group (23.1% vs. 56.57%, $p<0.001$), while combined ascending aortic and hemiarch replacements under hypothermic circulatory arrest conditions were performed more in the DTLC group (75.09% vs. 36.65%, $p<0.001$). As a result, CPB perfusion times were longer in the DTLC group (171±76 min vs. 156±88 min; $p = 0.047$) and mean core temperatures were lower (21±6˚C vs. 24.6±7˚C; $p<0.001$) compared to the non-DTLC group. On the other hand, aortic cross clamp times were similar in both groups (86±4 min vs. 89±34 minutes; $p = $ ns). Detailed surgical information for all patients and each subgroup is listed in Table 2.

## Operative outcome

There were no differences in the re-exploration for bleeding rates, requirements for renal replacement therapy and in-hospital stays between the two groups (Table 3). Neurological deficits were noted prior to surgery in 15.9% of patients in the DTLC group and in 11.6% of patients in the non-DTLC group ($p = 0.352$). New neurologic events were diagnosed in around 8% of patients in each group following surgery ($p = 0.983$), but there was a trend for quicker neurological recovery in the DTLC-group (54.5% vs. 41.37%; $p = 0.077$) (Table 4). In-hospital mortality rates and 30-day mortality rates did not differ between the two groups (10.1% vs. 11.1% and 11.55% vs. 11.95% respectively; both $p = $ ns) (Table 3).

**Table 2. Intra-operative and post-operative data.**

|  | DTLC group | Non-DTLC group | *p* |
|---|---|---|---|
| CPB time (min) | 171 ± 76 | 156 ± 88 | 0.047 |
| X-clamp time (min) | 86±40 | 89±34 | 0.076 |
| lowest temperature (˚C) | 21±6 | 24.6±6.5 | <0.001 |
| HCA | 210 (75.81%) | 99 (39.44%) | <0.001 |
| HCA time (min) | 22.98±9 | 25.2±11 | 0.07 |
| aCR | 27 (9.75%) | 19 (7.57%) | 0,376 |
| rCP | 14 (5.05%) | 9 (3.59%) | 0.409 |
| Extent of aortic replacement | | | |
| ascending aorta | 64 (23.1%) | 142 (56.57%) | <0.001 |
| ascending aorta + hemi-arch | 208 (75.09%) | 92 (36.65%) | <0.001 |
| ascending aorta + total arch | 5 (1.8%) | 17 (6.77%) | 0.067 |

Data is presented as mean±SD or as number (%) as appropriate.

*CPB*: cardiopulmonary bypass, *X-clamp time*: aortic cross-clamp time, *HCA*: hypothermic circulatory arrest, *aCR*: antegrade cerebral perfusion, *rCR*: retrograde cerebral perfusion.

Follow-up data were available for 98% of patients for a mean period of 4.3±3.3 years. The estimated 5-year and 10-year survival rates were similar in both groups (77.4% vs. 80.4% and 57.59% vs. 65.59% respectively; both *p* = 0.149) (Fig 2, Table 3).

## Discussion

While optimal arterial cannulation technique for CPB in A-AAD remains debatable, femoral arterial cannulation with retrograde CPB perfusion has been used extensively in clinical practice [1, 8]. Clinical studies have identified several drawbacks to the reversed arterial flow through the dissected aorta [9] including risk of retrograde embolization of thrombi from the false lumen [10, 11], aggravating the dissection through perfusing the false lumen [12] or inducing additional intimal injuries [13], or compression of the true lumen or its arterial branches, with potential peripheral or cerebral malperfusion [13–16]. Comparatively, antegrade perfusion has been shown to more effective in preventing malperfusion, embolization and organ dysfunction [2].

Several alternative techniques to femoral arterial cannulation have been used to achieve antegrade CPB perfusion in A-AAD patients. Axillary arterial cannulation can restore antegrade cerebral perfusion, but CPB flow may be insufficient if the vessel is narrow. Moreover, it

**Table 3. Patients' morbidity, mortality and survival.**

|  | DTLC group | Non-DTLC group | *p* |
|---|---|---|---|
| re-exploration for bleeding | 25 (9%) | 31 (12.35%) | 0.215 |
| CVVHD | 45 (16.25%) | 31 (12.35%) | 0.203 |
| in-hospital stay (days) | 13.09±13.18 | 12.87±15.87 | 0.4 |
| in-hospital deaths | 28 (10.1%) | 28 (11.1%) | 0.696 |
| 30-day mortality | 32 (11.55%) | 30 (11.95%) | 0.887 |
| 5-year survival | 77.4% | 80.4% | 0.149 |
| 10-year survival | 57.59% | 65.59% | 0.149 |

Data is presented as mean±SD or as number (%) as appropriate.

*CVVHD*: continuous veno-venous hemodialysis.

**Table 4. Patients' preoperative and postoperative neurological status.**

|  | DTLC group | Non-DTLC group | p |
|---|---|---|---|
| Pre-operative neurological status | | | |
| no pathology | 223 (80.5%) | 213 (84.9%) | 0.352 |
| not assessable | 10 (3.6%) | 9 (3.6%) | |
| positive deficits | 44 (15.9%) | 29 (11.6%) | |
| Post-operative neurological status | | | |
| new events | 19 (8.5%) | 17 (8.0%) | 0.983 |
| resolved events | 24 (54.5%) | 12 (41.37%) | 0.077 |

Data is presented as number (%).

may lead to arterial injury or dissection with possible cerebral or limb hypoperfusion [17], paradoxical arm hyperthermia [18], or brachial plexus or subclavian vein damage [15, 19, 20]. Carotid artery [15] or innominate artery cannulations [21] have been used, but are time consuming and may induce ipsilateral cerebral hyperperfusion. Left atrial cannulation or transapical left ventricular approach across the aortic valve into the true lumen have also been proposed, but are technically demanding and may lead to left ventricular dissection or life-threatening bleeding after decannulation [22, 23].

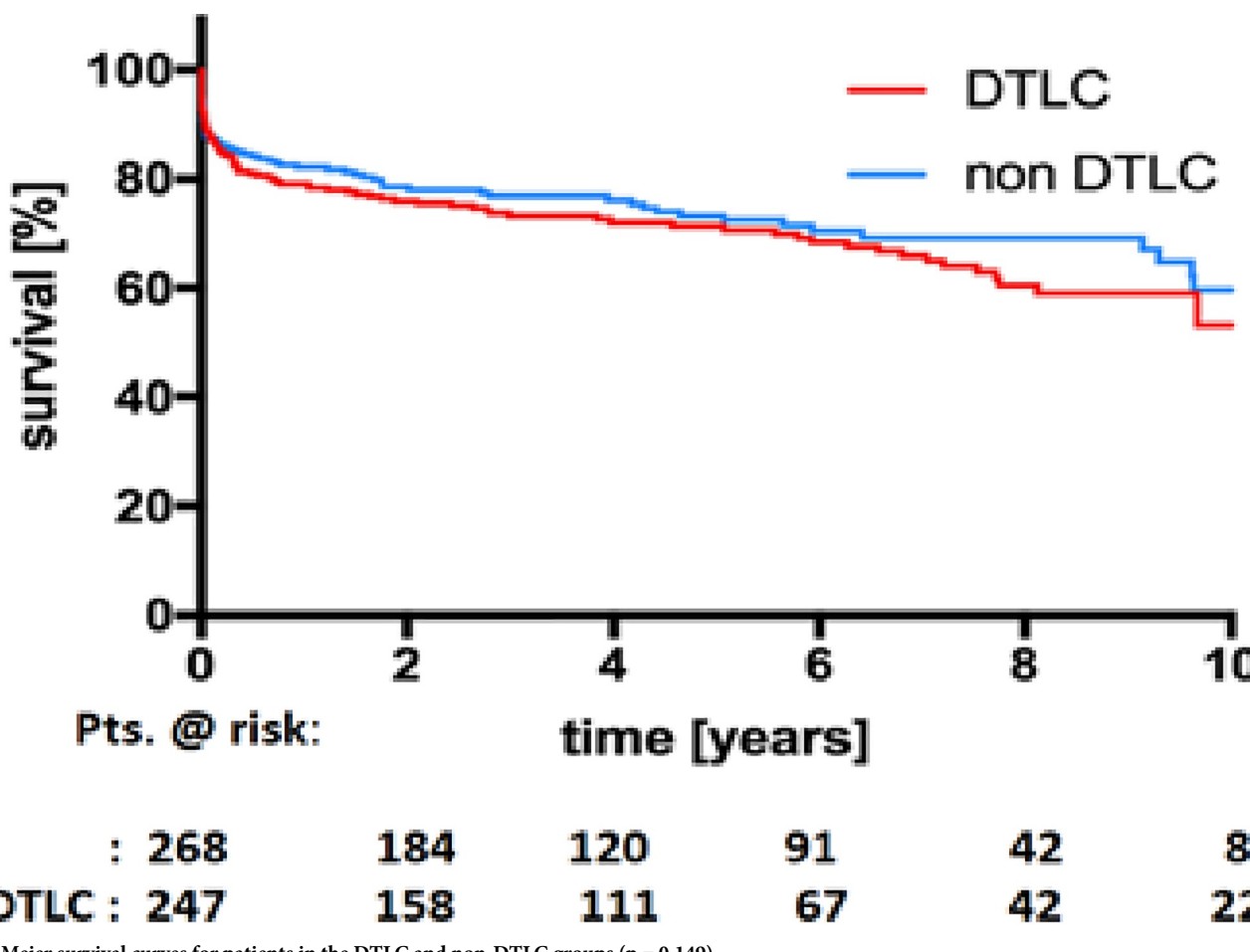

Pts. @ risk:

| | | | | | | |
|---|---|---|---|---|---|---|
| DTLC | : 268 | 184 | 120 | 91 | 42 | 8 |
| Non-DTLC | : 247 | 158 | 111 | 67 | 42 | 22 |

**Fig 2. Kaplan-Meier survival curves for patients in the DTLC and non-DTLC groups (p = 0.149).**

Direct true lumen cannulation to achieve antegrade CPB perfusion in A-AAD patients was first described by Borst *et al* [3]. His original technique involved initiating retrograde CPB perfusion via a peripheral arterial cannula, and then inserting a second arterial cannula into the proximal aorta if obstruction of the true lumen ensued [16]. Starting with ascending aortic cannulation in A-AAD patients was later described by Lijoi *et al* [24] and modified by Minatoya *et al* and others [4, 6, 25]. Methods to identify the true lumen included transesophageal echocardiography [26] or epiaortic ultrasound, but were time-consuming in emergency conditions or in the presence of circumferential aortic dissection or true lumen collapse [27]. Nevertheless, DTLC was shown to be safe and effective in patients with A-AAD [4, 6, 7, 28, 29], and restoring antegrade CPB perfusion was possible despite the complexity of the existing clinical scenarios, such as true lumen collapse, infrarenal aneurysm, peripheral vascular disease or obesity [6].

Benefits that can be attributed to antegrade perfusion provided by central aortic cannulation include prompt establishment of CPB after sternotomy, avoidance of further propagation of the dissection, prevention of retrograde cerebral embolization, avoidance of peripheral arterial injury, as well as enhanced neurological outcome [2, 8, 27]. In particular, DTLC may be specifically beneficial in patients with true lumen collapse since the antegrade perfusion will expand the true lumen and reduce hypoperfusion, while retrograde perfusion via the femoral artery may end up supplying blood flow to the false lumen, thus compressing the true lumen even more. As stated above, DTLC was our preferred strategy for patients in shock, tamponade or under resuscitation. Sternotomy in such extreme circumstances facilitates the immediate release tamponade and internal cardiac massage, together with establishing DTLC.

In this report, we compared the immediate, short-term and long-term outcomes of DTLC with alternative cannulation techniques in a large cohort of patients with A-AAD. Our two groups of patients were comparable to other publications according the Penn classification of preoperative clinical status [30–32]. Even though the preoperative clinical status according to the Penn classification was significantly worse in the DTLC group, the outcomes were similar. Still more, there was a trend for a faster resolution of preoperative neurological deficits in the DTLC group.

Despite the liberal use of DTLC, we did not encounter any incident of aortic rupture or any cannulation-related complications. We feel that it is necessary to avoid excessive introduction of the cannula into the aorta, and to divert the tip of the cannula away from the aortic wall to prevent inducing additional intimal injuries. The perfusionist must also confirm that there is no high pressure on the cannula prior to starting CPB perfusion.

In summary, our data shows DTLC to be a safe and effective cannulation strategy in patients with A-AAD under resuscitation, in shock or with dissection of the innominate or subclavian arteries. We believe that aortic surgeons should be familiar with various cannulation techniques, including DTLC, in order to be able to select the optimal cannulation strategy based on the preoperative clinical and anatomical specifics of each individual patient.

## Supporting information

**S1 Video. Direct true-lumen cannulation supplementary video.**
(MP4)

## Acknowledgments

Presented at the 32nd Annual Meeting of the European Association for Cardio-Thoracic Surgery, Milan, Italy, 18–20 October 2018

## Author Contributions

**Conceptualization:** Hazem El Beyrouti.

**Data curation:** Hazem El Beyrouti, Daniel-Sebastian Dohle, Philipp Pfeiffer.

**Formal analysis:** Hazem El Beyrouti.

**Investigation:** Hazem El Beyrouti, Lena Brendel.

**Methodology:** Hazem El Beyrouti.

**Project administration:** Hazem El Beyrouti.

**Software:** Hazem El Beyrouti.

**Supervision:** Hazem El Beyrouti.

**Validation:** Hazem El Beyrouti.

**Visualization:** Hazem El Beyrouti.

**Writing – original draft:** Hazem El Beyrouti.

**Writing – review & editing:** Hazem El Beyrouti, Daniel-Sebastian Dohle, Mohammad Bashar Izzat, Christian-Friedrich Vahl.

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
