## [Decision Letter · Decision Letter 0]

27 Aug 2020

PONE-D-20-21577

Direct true lumen cannulation in type A acute aortic dissection

PLOS ONE

Dear Dr. El Beyrouti,

Thank you for submitting your manuscript to PLOS ONE. After careful consideration, we feel that it has merit but does not fully meet PLOS ONE’s publication criteria as it currently stands. Therefore, we invite you to submit a revised version of the manuscript that addresses the points raised during the review process.

The reviewers have commented on your paper. They have suggested that this manuscript be revised according to the reviewers suggestions and resubmitted. Provided you address the changes recommended, the manuscript will be accepted for publication.

We look forward to receiving your revised manuscript.

Kind regards,

Prof. Raffaele Serra, M.D., Ph.D

Academic Editor

PLOS ONE

Journal Requirements:

2. Please ensure you have thoroughly discussed any potential limitations of this study within the Discussion section.

3. Thank you for including your ethics statement:  "Approval from our institutional ethics committee was obtained for data analysis (2018-13574-Epidemiologie). Mainz. Germany".   

Additional Editor Comments (if provided):

The reviewers have commented on your paper. They have suggested that this manuscript be revised according to the reviewers suggestions and resubmitted.

Reviewers' comments:

Reviewer's Responses to Questions

**Comments to the Author**

1. Is the manuscript technically sound, and do the data support the conclusions?

Reviewer #1: Yes

Reviewer #2: Yes

2. Has the statistical analysis been performed appropriately and rigorously? 

Reviewer #1: I Don't Know

Reviewer #2: Yes

3. Have the authors made all data underlying the findings in their manuscript fully available?

Reviewer #1: Yes

Reviewer #2: Yes

4. Is the manuscript presented in an intelligible fashion and written in standard English?

Reviewer #1: Yes

Reviewer #2: Yes

5. Review Comments to the Author

Reviewer #1: 528 patients with Type A aortic Dissection (2006-2017) where a unique Direct True Lumen cannulation technique was employed n 52.4% of their patients with similar outcomes (which are acceptable ~30 day mortality 12%, 8% new neuro deficits) to their conventional peripheral cannulation techniques (same).

Their novel direct technique involves encircling the dissected aorta (dangerous) and a brief normothermic drop in temperature (also dangerous) and dividing the dissected aorta to place the aortic cannula under direct vision (more dangerous) and would be best reserved for experienced surgeons.

At our center and at Penn, direct cannulation is performed using a “Seldinger” wire technique under epi-aortic ultrasound and TEE guidance, which is much safer (in my opinion) with similar outcomes in obtaining true lumen cannulation and avoiding the pitfalls of peripheral cannulation.

For this article to be published, The discussion should include references to the Seldinger technique and specific reasons why the Mainz center group feels their approach is preferred.

Furthermore, if published the manuscript would benefit from figures showing their preferred technique.

Reviewer #2: Thanks for your nice report.

Another article about femoral cannulation that you might cite is Fusco, DS. Ann Thorac Surg. 2004 Oct;78(4):1285-9; discussion 1285-9.

Your paper could be enhanced by a nice accompanying drawing of your cannulation technique.

Although you demonstrated no benefit from this arterial cannulation technique over other methods, the description of your technique is still valuable.

How often did you have troubling bleeding from around the cannula during perfusion?

Did your surrounding aortic tapes ever tear through the fragile aortic layers.

Among the other cannulation options, you mention "left atrial". Please explain.

6. PLOS authors have the option to publish the peer review history of their article (what does this mean?). If published, this will include your full peer review and any attached files.

Reviewer #1: No

Reviewer #2: **Yes: **John A. Elefteriades, MD

---

## [Author Response · Author response to Decision Letter 0]

17 Sep 2020

Responses to Reviewer #1

1) We believe that antegrade perfusion by central aorta cannulation prevents retrograde cerebral embolization and avoids peripheral arterial injury. As mentioned in sentence number 207-209, cannulation using the Seldinger technique can be time-consuming in emergency cases and may be challenging in the presence of circumferential aortic dissection accompanied with central true lumen collapse.

2) New reference Penn ascending cannulation using the Seldinger technique (ref. no. 23. Schoeneich et al) was added.

3) A video is added to demonstrate the operative technique. Also added is another reference from our department detailing the technique (ref. no. 7. Conzelmann et al). 

Responses to Reviewer #2

1) The reference (ref. no. 8. Fusco et al) was added as requested.

2) A video is added to demonstrate the operative technique. Also added is another reference from our department detailing the technique (ref. no. 7. Conzelmann et al). 

3) In our early experience, after careful dissection between the adventitia of the ascending aorta and the pulmonary artery, we used to place one Mersilene tape around the fragile aorta (ref. no. 7. Conzelmann et al). However, we occasionally encountered some bleeding from around the cannula during CPB perfusion. We now use double Mersilene tape which provide better stability and prevent bleeding from around the aortic cannula. 

4) Fortunately, we never had any episode of aortic rupture or tear due to Mersilene tapes in our experience. 

5) The left atrial cannulation Reference was added (ref no. 23. Schoeneich et al).

---

## [Editor Report · Decision Letter 1]

21 Sep 2020

Direct true lumen cannulation in type A acute aortic dissection: A review of an 11 years’ experience

PONE-D-20-21577R1

Dear Dr. El Beyrouti,

We’re pleased to inform you that your manuscript has been judged scientifically suitable for publication and will be formally accepted for publication once it meets all outstanding technical requirements.

Kind regards,

Prof. Raffaele Serra, M.D., Ph.D

Academic Editor

PLOS ONE

Additional Editor Comments (optional):

amended manuscript is acceptable
---

## [Editor Report · Acceptance letter]

2 Oct 2020

PONE-D-20-21577R1 

Direct true lumen cannulation in type A acute aortic dissection: A review of an 11 years’ experience 

Dear Dr. El Beyrouti:

I'm pleased to inform you that your manuscript has been deemed suitable for publication in PLOS ONE. Congratulations! Your manuscript is now with our production department. 

Kind regards, 

on behalf of

Prof. Raffaele Serra 

Academic Editor

PLOS ONE